# Analysis and Improvement Measures of Driving Range Attenuation of Electric Vehicles in Winter

**Shuoyuan Mao** [1], **Meilin Han** [1], **Xuebing Han** [1], **Jie Shao** [2], **Yong Lu** [3], **Languang Lu** [1,*] **and Minggao Ouyang** [1,*]

1   State Key Laboratory of Automotive Safety and Energy, Tsinghua University, Beijing 100084, China; mao_sy@126.com (S.M.); hml20@mails.tsinghua.edu.cn (M.H.); hanxuebing@tsinghua.edu.cn (X.H.)
2   SAIC GM Wuling Automobile Co., Ltd., Liuzhou 545007, China; jie.shao@sgmw.com.cn
3   Beijing KeyPower Technologies Co., Ltd., Beijing 100015, China; yong.lu@key-power.com.cn
*   Correspondence: lulg@tsinghua.edu.cn (L.L.); ouymg@mail.tsinghua.edu.cn (M.O.); Tel.: +86-136-8313-0967 (ext. 100084) (L.L.); +86-139-1112-3499 (ext. 100084) (M.O.)

**Abstract:** A great many EVs in cold areas suffer from range attenuation in winter, which causes driver anxiety concerning the driving range, representing a hot topic. Many researchers have analyzed the reasons for range attenuation but the coupling mechanism of the battery as well as the vehicle and driving conditions have not been clearly estimated. To quantitatively investigate the driving range attenuation of electric vehicles (EVs) during winter, an EV model mainly integrated with a passenger-cabin thermal model, battery model, and vehicle dynamic model was constructed and simulated based on the mass-produced Wuling HongGuang Mini EV. Real vehicle dynamic driving data was used to validate the model. Based on NEDC driving conditions, the driving range calculation formula and energy flow diagram analysis method were used. The reason for attenuation was evaluated quantitatively. Results show that battery energy loss and breaking recovery energy loss contribute nearly half of the range attenuation, which may be alleviated by battery preheating. Suggestions for extending driving range are proposed based on the research.

**Keywords:** electric vehicle; driving range; temperature effects; battery heating strategies

## 1. Introduction

The proportion of electric vehicle (EV) production and sales volume are increasing around the world. During winter, many EVs in areas with temperatures below zero suffer from the problem of range attenuation and cannot even be driven at all [1]. An earlier on-road test on Chevrolet Volt showed that the driving range at low temperatures would decrease to 47.5% of the range at normal temperatures [2]. The winter problem is one of the main reasons that restrict the entry of electric vehicles into the market.

There are already many researchers that focus on the performance attenuation of li-ion batteries at low temperatures. The internal resistance of the lithium battery increased significantly at low temperatures, resulting in a significant decrease in the operating voltage, capacity, power, and energy. In addition, charging at low temperatures can easily lead to lithium precipitation at the negative surface, which can cause permanent battery damage and may cause serious safety problems [3]. It is generally accepted that the reasons for the poor performance of lithium ion batteries at low ambient temperatures are: low electrolyte conductivity; increased resistance to solid electrolyte interface membranes (SEI); slow diffusion of lithium in SEI membranes and active material particles; and slow charge transfer dynamics [4–6]. The speed control steps are different at different discharge multiples and at different batteries' self-producing heat cases. Regardless of the battery self-heating, charge transfer dynamics is the limiting factor at a very small discharge rate. As the multiplier increases, the main limiting factor becomes the increase of electrolyte resistance. At low temperatures, the heat generation of batteries increases due to the increase in ohmic internal resistance, which causes a significant temperature rise when

discharging at a medium power; then, the concentration polarization decreases and the resistance of negative solid particles becomes the limiting factor of voltage and capacity.

The performance of batteries at low temperatures not only depends on the ambient temperature but also on the discharge multiple and thermodynamic conditions [7]; the capacity is lower when the battery is charged/discharged at a higher C rate and it can be improved by internal heating, such as alternative current heating, direct current heating, or external heating, which mainly refers to electric heating, hot fluid heating, or heating using phase change materials [8]. According to the energy source, the battery heating method can be classified into heating with energy from the battery itself and heating with external energy. According to the source of heating production, it can be classified into the battery's joule heat and external heating device. Based on the Joule heat generation of the battery's internal resistance, a battery structure with an embedded nickel chip was designed, which can carry out high-power self-heating [9,10]. For battery heating at extreme low temperatures, an aluminum heating sheet was bonded between two batteries to generate heat with the energy of the battery itself [11]; while the battery is heated, the joint state estimation is performed to estimate the maximum discharge current by using the multiple constraint strategy to avoid over-discharge. Another study used a resonant high-frequency oscillator to generate high-frequency AC currents based on LC oscillation to heat the battery with the its own energy and the structure achieved both high speed and high efficiency [12]. Scientists also established a method using a combination of the heat generated by the battery itself and the MOSFET to heat the battery, and the safety is ensured by controlling the current through using the conduction characteristics of the MOSFET, which achieves safe and rapid heating [13]. In addition, the constraints on safety and durability should be considered when designing the battery heating strategy [14–16]. Additionally, a study on the capacity fading effect of low-rate charging on lithium-ion batteries in low-temperature environment showed that the main mechanism of the aging of the battery is the loss of active lithium under the conditions of low-rate cycling at sub-zero temperatures [17]. To accurately analyze the performance of electric vehicles in various conditions, some vehicle models were established to conduct the simulation [18].

The research on battery low temperature and heating was reviewed above but there are few studies that focus on the low temperature performance and effect of heating on driving range at the vehicle level. A study analyzed the impact of two factors separately on a battery electric vehicle's driving range and the results showed that the range can decrease by up to 31.9% due to heating and by up to 21.7% due to limited recuperation, which gives a combined maximum range decrease of approximately 50% under cold conditions [19]. The reasons for range attenuation at low temperatures, however, have not been fully understood; the coupling mechanism of the battery as well as the vehicle and driving conditions have not been clearly estimated; and the effect of heating on the improvement of endurance is still not clear.

In this paper, based on the micro-EV Mini (whose pack is composed of 32 LiFePO4 prismatic cells in series with a capacity of 135 Ah), an EV model is constructed and simulated. The thermal model of the passenger cabin, the battery model, and the vehicle dynamic model are combined together to build the vehicle energy flow model. Real vehicle driving data is used to validate the model and the model output agrees well with dynamic vehicle data. Based on NEDC driving conditions, the reason for the range attenuation is evaluated quantitatively. Results show that the greatest part of the energy consumption that causes range attenuation is the air conditioner energy consumption. Additionally, battery energy loss and breaking recovery energy loss due to low temperatures contribute nearly half of the range attenuation, which are caused by the battery characteristic at low temperatures and can be alleviated by battery preheating. Then, extensive simulations are carried out and the optimal heating method for increasing driving range is proposed. We found that preheating the battery can significantly improve driving range at low SOC in cold weather; preheating the battery to 5 °C is also a good strategy considering both energy saving and performance. Moreover, the heating strategy is made in consideration

of velocity, environment temperature, and initial SOC. Finally, conclusions are made and suggestions for extending driving range are proposed.

The paper is organized as follows. The experimental platform and device are explained in Section 2. The model of Wuling Hongguang Mini EV is described in detail in Section 3. The model is validated with on-road driving data in Section 4. The simulation method and results are described in Section 5. Conclusions and suggestions for improvement are made in Section 6.

## 2. The Configuration

The experimental device in this study includes a bench test and hardware in the loop, as shown in Figure 1. In the bench test part, the XinWei CT-8000 battery test system was used to monitor the battery charging and discharging steps; the Bell BE-THK incubator was used to control the experimental temperature; and the K-type thermocouple was used to cooperate with the Pico TC-08 thermocouple data recorder for temperature measurement. The accuracy of the thermocouple was 1.5 °C. In the hardware in the loop part, a KeyPower BMS was used to monitor the voltage, current, and temperature of the assembly module, and to estimate the battery state. Eight thermocouples were used to correctly measure the battery's temperature during the identification of the battery parameters; six were placed in the center of each outer surface of the battery and two were implanted into the battery. The position of the two inner thermocouples is shown in Figure 2a. Another eight thermocouples were used to measure the temperature changes of the battery bulk and all of the sensors were attached to the electrode tab; the position is shown in Figure 2b, representing the top view of the battery pack. Each brown block represents a battery and the red points represent the thermocouples.

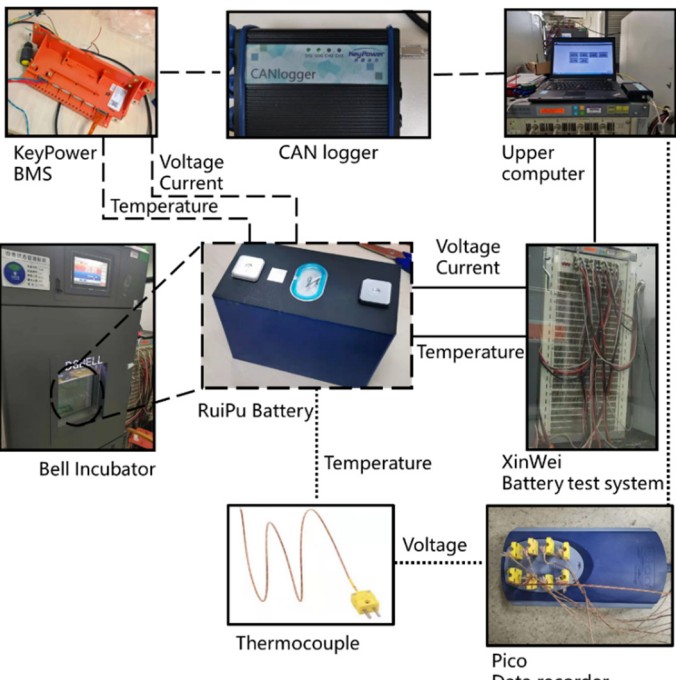

**Figure 1.** Experimental device.

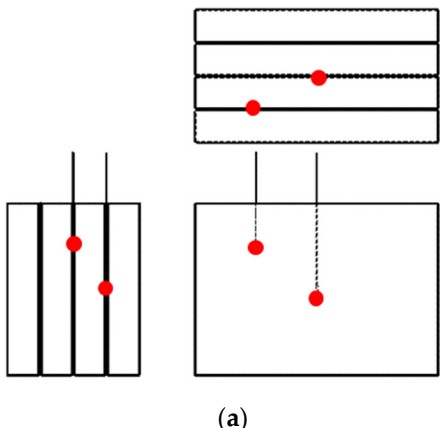 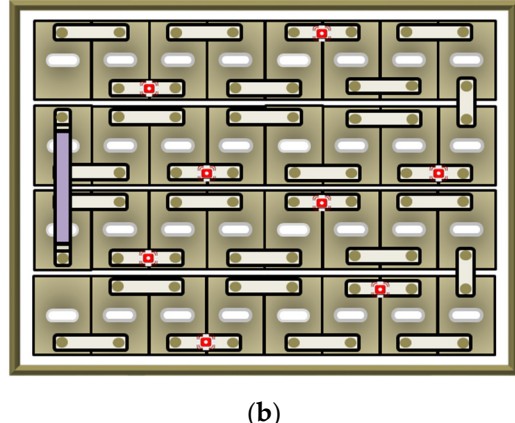

　　　　　　　　(**a**)　　　　　　　　　　　　　　　　　　　　(**b**)

**Figure 2.** Position of the thermocouples: (**a**) position of two thermocouples put into the battery and (**b**) position of eight thermocouples to measure the battery bulk temperature.

The research object of this study is the Wuling HongGuang Mini EV, which is one of the best-selling electric vehicles in China. It is equipped with a power battery system supplied by Beijing KeyPower Technology Co., Ltd. (Beijing, China), and the whole battery pack is composed of 32 prismatic batteries in series. The capacity of the battery is 135 Ah and the cathode material is phosphate iron lithium.

## 3. Modeling of the Wuling HongGuang Mini EV

### 3.1. Thermal Model of the Passenger Cabin

To model the passenger cabin correctly, parameters must be measured exactly. Structural parameters are mainly obtained through the information provided by its manufacturer and the actual measurement, and thermal parameters are determined through experience and the thermal formula.

Assuming the cabin is in an equilibrium thermodynamic state, the heat generation of the air conditioner ($Q_H$) equals to the heat dissipation of the passenger cabin, including the heat dissipation power of the panels in all directions ($Q_{panel}$), heat dissipation power of windows ($Q_{glass}$), heat generation power of passengers ($Q_{personnel}$), ventilation power ($Q_{air}$), and deforest wind power ($Q_{fog}$); the relation can be expressed by Equation (1) [20,21]:

$$Q_H = Q_{panel} + Q_{glass} + Q_{personnel} + Q_{air} + Q_{fog} \tag{1}$$

$$Q_{panel} = (T_{cabin} - T_{envi}) \cdot \left( F_{cabin\_front} K_{cabin\_front} + F_{cabin\_rear} K_{cabin\_rear} + F_{cabin\_side} K_{cabin\_side} + F_{cabin\_floor} K_{cabin\_floor} \right) \tag{2}$$

$$Q_{glass} = (T_{cabin} - T_{envi}) \cdot \left( F_{glass\_front} K_{glass\_front} + F_{glass\_rear} K_{glass\_rear} + F_{glass\_side} K_{glass\_side} \right) \tag{3}$$

Equations (2) and (3) are the calculation formulas of the heat dissipation power of the panels and the heat dissipation power of the windows, where $T_{cabin}$ is the temperature of the passenger cabin in $K$; T_envi is the temperature of the external environment in $K$; $F$ is the area of each part in m$^2$ and the subscript indicates its location; and $K$ is the heat transfer coefficient in W/(m$^2$·K), which are functions of the vehicle speed and are calculated through empirical tests. The subscript indicates its location; for example, the subscript cabin_front represents the front of the cabin, while the subscript glass_front represents the front window.

With the thermal model of the passenger cabin, the relationship between the air conditioning power and the temperature inside the passenger cabin can be simulated.

*3.2. Battery Model*

The battery model includes the battery electrical model and the battery pack thermal model. The battery electrical model can be used to simulate the change of SOC and the relationship between its power, current, and voltage. The battery pack thermal model is modelled to calculate the temperature rise of the battery while continuously discharging. These two sub-models are coupled through SOC and the temperature, and can together simulate the thermoelectric state of the pack.

### 3.2.1. Battery Parameters

Parameters for modelling the electric model mainly include the resistance–SOC curve and the open-circuit voltage (OCV)–SOC curve both under different temperatures. To obtain these curves, HPPC tests were done under different SOCs and temperatures, and resistance was identified with the least square method. The HPPC test is a battery test cycle specified in the America FreedomCAR Battery Test Manual For Power-assist Hybrid Electric Vehicles. Charge and discharge plus current was applied in a specific SOC point and the battery parameters can be identified by observing the voltage response. After each HPPC test, a rest for 3 h was ensured and the last point of the voltage curve was identified as the OCV. Figure 3 shows these two maps.

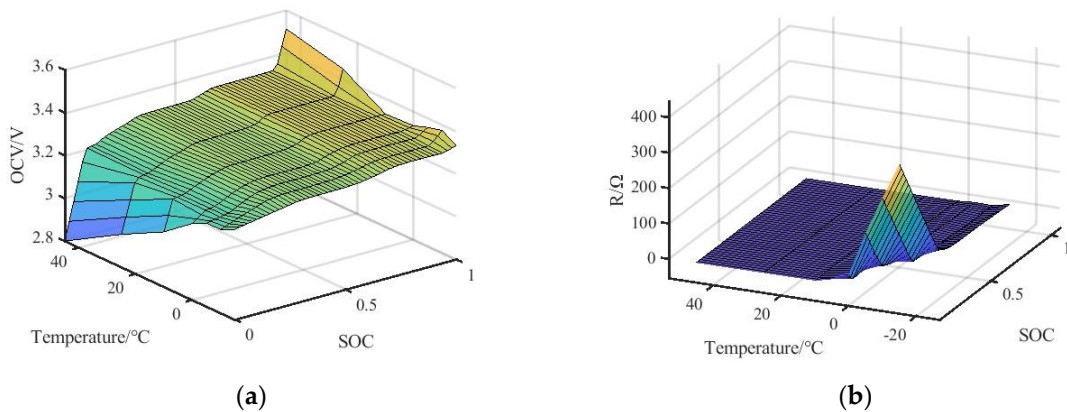

**Figure 3.** Parameter maps: (**a**) open-circuit voltage and (**b**) resistance.

Parameters for modelling the thermal model include mass and capacity of a single cell, size/mass and thermal parameters of the battery pack, location and power of the positive temperature coefficient (PTC) heating plate, which are given by the battery supplier.

### 3.2.2. Battery Equivalent Circuit Model

In this section, for simplicity, we use the Rint model to calculate terminal voltage, which is shown as the following equation:

$$U_t = U_{ocv} - IR_{in} \tag{4}$$

where $U_t$ is the terminal voltage of the battery in V; $U_{ocv}$ is the open-circuit voltage of the battery in V; $I$ is the current of the battery in A; and $R_{in}$ is the internal resistance of the battery in Ω.

To calculate SOC, the coulomb counting method was used, as depicted in Equation (5):

$$SOC = SOC_0 - \frac{\int_0^t \eta I d\tau}{3600C_{cap}} \tag{5}$$

where $SOC_0$ is the initial *SOC* in percentage; $\eta$ is coulomb efficiency; and $C_{cap}$ is the capacity of the battery in Ah. Moreover, the capacity used in the calculation of SOC is taken as a reference value, which is tested at 25 °C and 1/3C; it is a constant value and thus

not affected by the temperature and discharge rate when calculating SOC. This makes the SOC calculated as the standard SOC. Additionally, the standard SOC is also used when conducting battery tests and constructing look-up tables, as depicted in Figure 3, and this is the method to accurately derive parameters in various working conditions. The remaining driving range was used directly to define the energy condition of the battery in this research study, which will be discussed in Section 5.

### 3.2.3. Calculation of the Battery Heat Generation

Reference [22] simplified the heat production model, which is widely used to model the heat generation of battery systems. As shown in Equation (6), battery heat consists of reversible heat and irreversible heat (also called Joule heat). Irreversible heat is calculated by $I^2 R_{in}$ and is always positive regardless of the charge or discharge process. Reversible heat is calculated via the second term of Equation (6) and the sign of reversible heat is contrary during the charge and discharge process. Usually, irreversible heat is far larger than reversible heat, thus in many research studies, for simplicity, only irreversible heat is considered [23]. Thus, in this paper, the heat production is calculated via Equation (7):

$$\dot{Q}_{bat} = I^2 R_{in} - IT\frac{\partial OCV}{\partial T} \tag{6}$$

$$\dot{Q}_{bat} = I^2 R_{in} \tag{7}$$

where $\dot{Q}_{bat}$ is the heat production power in W.

### 3.2.4. Calculation of the Battery Pack Heat Dissipation

Assume that the battery pack is a zero-dimensional thermal model and the heat dissipation of the pack can be calculated with the same method of the passenger cabin. The author assumes the heat dissipation path begins from the cell and continues to the air inside the battery pack and then to the external environment.

$$\dot{Q}_{cell2air\_in} = F_{cell} \times K_{cell2air\_in} \times (T_{cell} - T_{air\_in}) \tag{8}$$

$$\dot{Q}_{air\_in2out} = (T_{air\_in} - T_{envi})(F_{side2envi}K_{side2envi} + F_{top2envi}K_{top2envi} + F_{bottom2envi}K_{bottom2envi}) \tag{9}$$

$$K = \frac{1}{\frac{1}{K\_in\_case} + \frac{\delta_{case}}{\lambda_{case}} + \frac{1}{1.63 \times (12\sqrt{v}+4)}} \tag{10}$$

In Equation (9), $\dot{Q}_{cell2air\_in}$ is the heat dissipation power from the cells to the air inside the battery pack in W; $\dot{Q}_{air\_in2out}$ is the heat dissipation power from the air inside the battery pack to the outer environment in W; $T$ is the temperature of each part in K; $F$ is the area of each part represented by the subscript in $m^2$; $K$ is the heat transfer coefficient between two parts represented by the subscript in $W/(m^2 \cdot K)$, as shown in Equation (10); $K\_in\_case$ is the convective heat transfer coefficient between the pack and environment; $\delta_{case}$ is the thickness of the battery aluminum housing; $\lambda_{case}$ is the thermal conductivity of the aluminum housing; and $v$ is the vehicle speed. This expression is an empirical formula to calculate the heat transfer coefficient.

### 3.2.5. Thermal Model of the Battery Pack

Considering heat generation and dissipation, the temperature change of cells can be calculated by the following differential equations:

$$\rho_{air\_in}V_{air\_in}c_{air\_in}\frac{dT_{air\_in}}{dt} = \dot{Q}_{cell2air\_in} - \dot{Q}_{air\_in2out} \tag{11}$$

$$m_{bat}c_p \frac{dT_{bat}}{dt} = \dot{Q}_{bat} + \dot{Q}_{heat} - \dot{Q}_{cell2air\_in} \tag{12}$$

where $\dot{Q}_{heat}$ is the pack-heating power with a heating plate in W; $\rho_{air\_in}$ is the air density inside the pack in kg/$m^3$; $V_{air\_in}$ is the air volume inside the pack in $m^3$; $c_{air\_in}$ is the specific heat capacity of the air inside the pack; $T_{air\_in}$ is the air temperature inside the battery pack; $m_{bat}$ is the mass of the battery pack; $c_p$ is the specific heat capacity of the pack; and $T_{bat}$ is the temperature of the pack.

### 3.3. Vehicle Dynamic Model

The vehicle dynamic model can provide the power demand towards the target speed. This is exactly the power output of the motor (with the transmission). With the motor's efficiency map, the motor's power input can be obtained, which is the power output of the battery pack. Then, the voltage and current can be calculated by the preciously discussed battery model. Parameters for the vehicle dynamic model are either measured or given by the manufacturer.

Assuming the load is perfect, specifically wherein the air speed is zero and the road is made from cement, resistance to be overcome by the drive motor is:

$$F_t = F_f + F_w + F_i + F_j \tag{13}$$

$$F_f = f \cdot mg \cdot cos\alpha \tag{14}$$

$$F_w = \frac{C_d}{2} A\rho u_a^2 \tag{15}$$

$$F_i = mg \cdot sin\alpha \tag{16}$$

$$F_j = m\frac{du_a}{dt} \tag{17}$$

where $F_t$ is the running resistance in N; $F_f$ is the rolling resistance in N; $F_w$ is the wind resistance in N; $F_i$ is the slope resistance in N; $F_j$ is the acceleration resistance in N; $m$ is the vehicle mass in kg; $g = 9.81$ in m/$s^2$; $\alpha$ is the slope angle; $C_d$ is the vehicle aerodynamic drag coefficient; $A$ is the windward area in m$^2$; $\rho$ is the external air density; and $u_a$ is the vehicle speed in m/s.

The drive motor's traction power is shown in the following equation:

$$P_e = \frac{F_t u_a}{3.6\eta_t} \tag{18}$$

where $P_e$ is the traction power of the driving motor in kW and $\eta_t$ is the driving system efficiency.

The efficiency map of the vehicle's motor/inverter is shown in Figure 4, wherein motor/inverter efficiency is determined by its speed and torque. Afterwards, the battery pack's power output can be calculated with the following equation:

$$P_b = \frac{P_e}{\eta_m} \tag{19}$$

where $P_b$ is the power output from the battery pack in kW and $\eta_m$ is the motor efficiency.

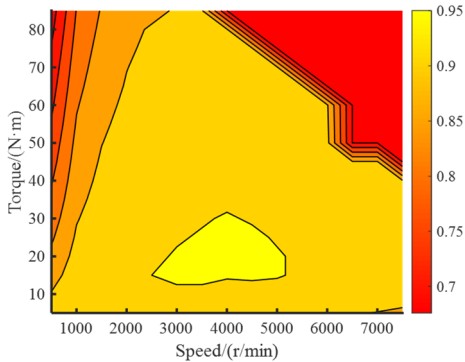

**Figure 4.** Efficiency map of the motor/inverter.

When the vehicle is breaking and energy is being recovered, the energy recovery power of the battery pack can be calculated via the following equation:

$$P_b = P_e \cdot \eta_m \tag{20}$$

From the perspective of the battery pack, the power output/recovered power can be depicted by the following equation:

$$P_e = U_t \cdot I \tag{21}$$

Hence, the current $I$ can be calculated. Then, the temperature rise and SOC change are obtained.

## 4. Model Validation

In this section, the vehicle model is constructed and both on-road driving data from the winter of 2020 and the Beijing of a Wuling HongGuang Mini EV are used to validate the model.

The logic diagram of modelling the vehicle is shown in Figure 5. Place the target speed and real speed into the driver model (a PID control model); the output is the accelerate/brake signal. The controller model can read the signal and output motor torque command. With torque and speed, the power to drive the vehicle is calculated, which is equal to the power output of the battery model. The current of the battery is then calculated with its power and voltage. Finally, the heat generation power is obtained. Then, the calculation proceeds to the next step.

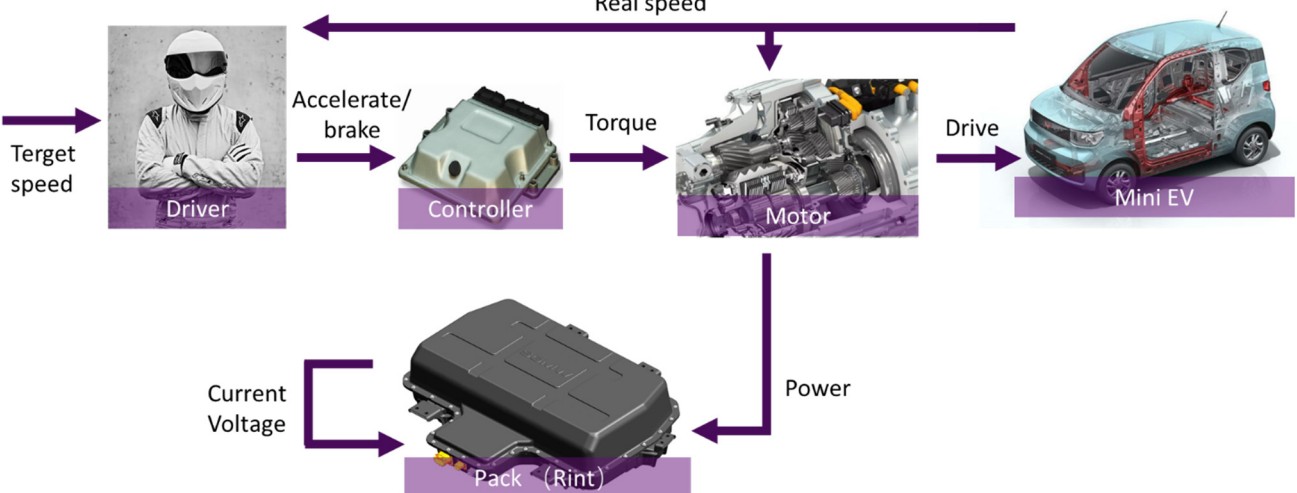

**Figure 5.** The simulation block diagram.

Two vehicles' real driving data were used to conduct the validation through the factory authorization, wherein both contained one month's worth of driving information during the winter. A database was set based on that. A continuous driving dataset was selected from the database and the basic information is listed in Table 1. The detailed data is shown in Figure 6 in blue.

**Table 1.** Basic information of the data to validate the model.

| Condition | Value |
|---|---|
| Environment temperature | −8 °C |
| Battery pack initial temperature | 5 °C |
| SOC change during driving | 100%–43% |
| Data and time | 10:00, 29 December 2020 |

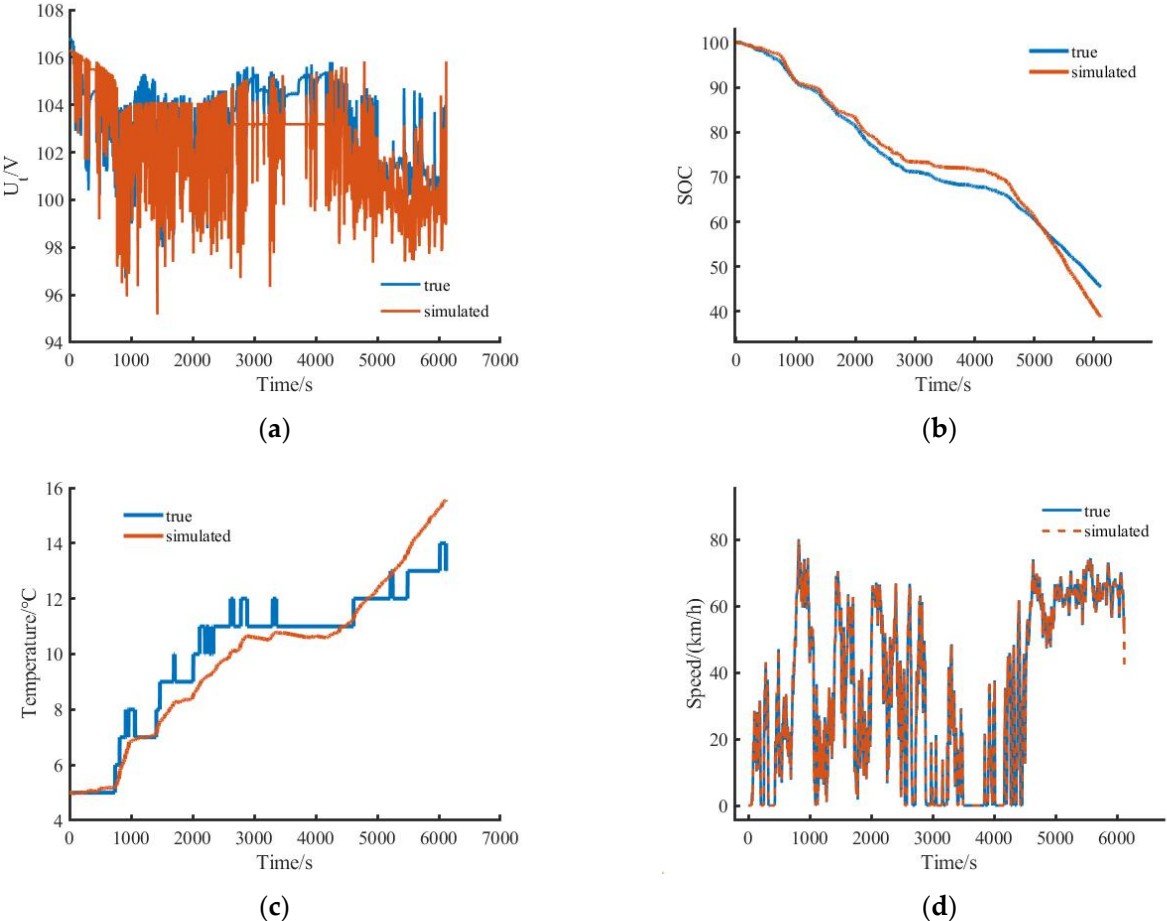

**Figure 6.** Validating the model and simulation results: (**a**) voltage; (**b**) SOC; (**c**) temperature; and (**d**) speed.

Set the environment parameters to that of 28 December 2020 and then put the target speed (true speed) into the model. The model can simulate a driver driving at the target speed and can calculate the current, SOC, voltage, temperature, power of each part, etc. The selected representative simulation results were compared with the reality and are shown in Figure 6 in red. Table 2 lists the root-mean square error (RMSE) of each result.

**Table 2.** RMSE of the simulation results.

| Important Physical Quantities | RMSE |
|---|---|
| Battery pack voltage | 1.55 V |
| Battery pack SOC | 2.58% |
| Battery pack temperature | 0.90 °C |
| Vehicle speed | 2.53 km/h |

The RMSE between the simulated value and the real vehicle driving data is all within 10% of the real data's range. Hence, this model's accuracy is considered enough for further simulation and analysis.

## 5. Results and Discussion

*5.1. Quantitative Analysis on the Reason for Range Attenuation during the Winter*

The energy float diagram is a diagram that shows the flow of energy, while the energy sources and flow path can be identified quantitively through the energy float diagram.

The driving range attenuation is estimated under the NEDC driving condition from 100% SOC to 0% SOC. The reason for attenuation is analyzed quantitatively using the energy flow diagram analysis method shown in Figure 7.

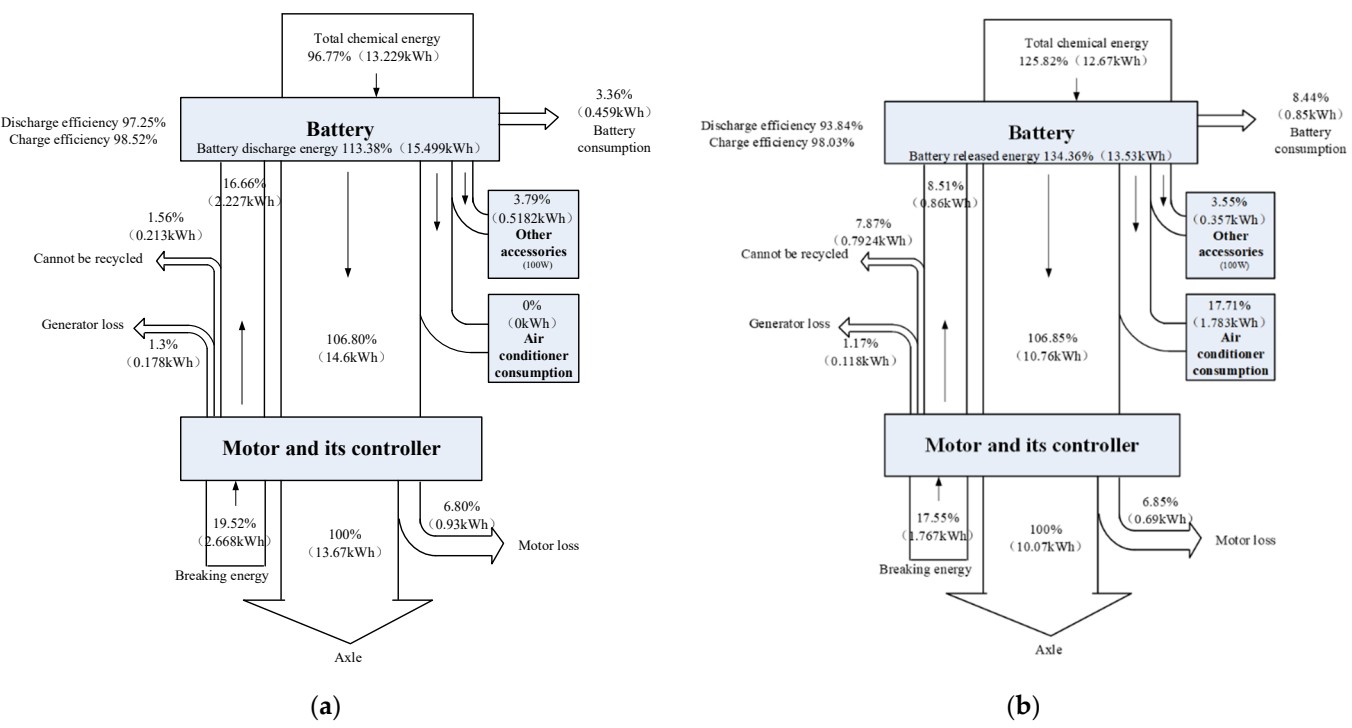

**Figure 7.** Energy float diagram under two conditions: (**a**) ambient temperature is 25 °C, battery pack initial temperature is 25 °C, and air conditioner is off; (**b**) ambient temperature is −10 °C, battery pack initial temperature is −10 °C, and air conditioner is on and maintains the cabin at 16 °C.

By comparing the two simulations above, different factors contributing to range attenuation were found, as shown in Figure 8. Some basic conclusions include: (1) energy consumption of the air conditioner accounts for a large proportion of energy consumption; (2) battery energy loss and breaking recovery energy loss due to low temperatures contribute nearly half of the range attenuation, which may be alleviated by battery preheating; and (3) low temperature causes an increase of the driving resistance.

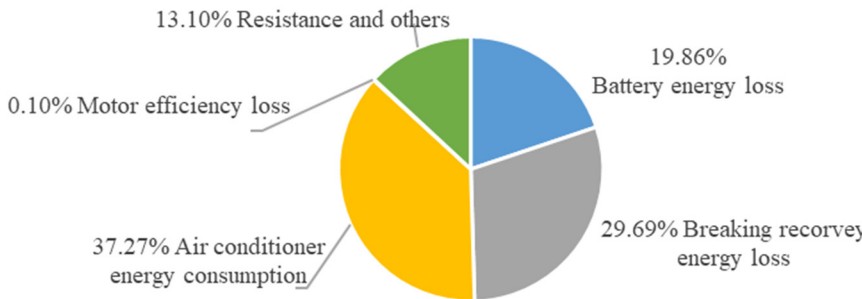

**Figure 8.** Pie chart of different factors' contribution to range attenuation.

Due to the large proportion of air conditioning energy consumption, we turned off the air conditioning switch in the following simulations in order to maximize the vehicle's potential.

*5.2. Comparative Analysis of Heating Strategies at Low Temperatures*

To evaluate the performance of different preheating temperatures, the next group of simulations was the driving range under different ambient temperatures, initial SOCs, and preheating temperatures. In each set of simulations, the ambient temperature and initial SOC were fixed and different initial temperatures of the battery pack were set (assuming that 1 degree of temperature rising consumes 0.15% SOC through heating). Both the driving range and the sum of the driving range under different initial SOCs were compared in order to assess the effects of each preheating temperature.

To find the appropriate preheating temperature, the model preheating temperature increased by every 5 °C in the simulation. At each ambient temperature, the model was simulated at every 10% initial SOC from 10% SOC to 100% SOC. To find a general optimum result, the driving range under different SOCs was summed, as shown in Figure 9a. Each line represents an average driving range at the same ambient temperature, wherein the preheat temperature of each point on the line is different. Regardless of the ambient temperature, preheating the battery to 5 °C before driving can reach the best performance.

More details are given in Figure 9b–e. The vehicle driving range with preheating and without preheating were compared under different initial SOC conditions. The common conclusion could be made: preheating the battery can significantly improve driving range at low SOC in cold weather and preheating the battery to 5 °C is a good strategy in consideration of both energy saving and performance.

An "anomalous" behavior was found according to the simulation. The results of the simulation for different ambient temperatures tended to show a common trend with unexpected lowering at the preheating temperature of 10 °C, especially for the ambient temperature of −15 °C. This phenomenon may lead to an unknown characteristic of the battery. It may be caused by the discharge characteristics (resistance), which is marginal not good. The energy recovery minus heating cost represents a marginal decline. Additionally, this anomalous phenomenon has a relationship with the driving protocol. To fully understand the phenomenon, future research should be conducted to go deep into the material.

To verify the applicability of the heating strategy, another group of simulations was set under the NEDC cycle and the simulation method was same as the former. Compared to the NEDC cycle, the WLTC cycle is more radical and represents the driving characteristic of Chinese urban roads. The results are given in Figure 10, indicating that preheating the battery to 5 °C is still a good strategy.

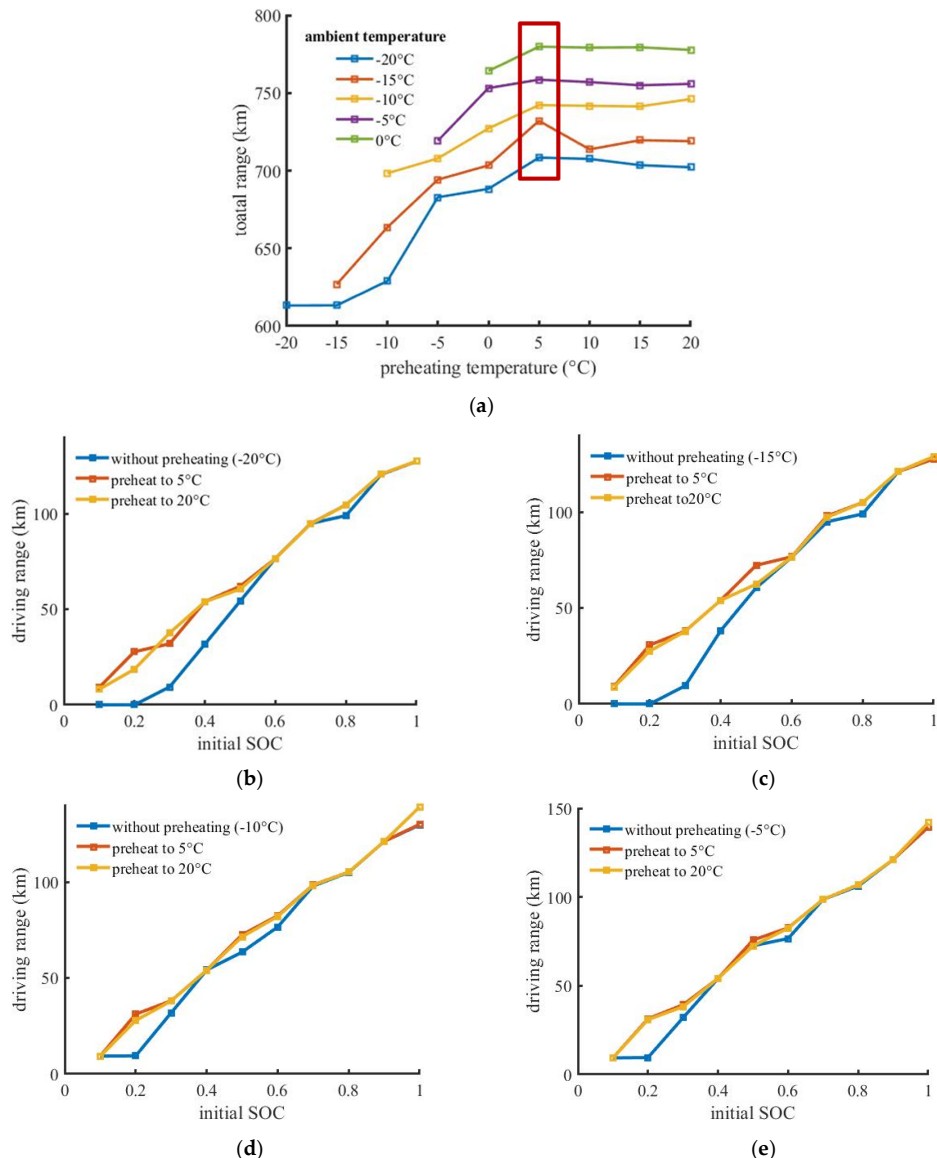

**Figure 9.** Simulation results in different ambient temperatures under the WLTC driving condition: (**a**) total driving range; (**b**) detailed results of the ambient temperature of −20 °C; (**c**) detailed results of the ambient temperature of −15 °C; (**d**) detailed results of the ambient temperature of −10 °C; and (**e**) detailed results of the ambient temperature of −5 °C.

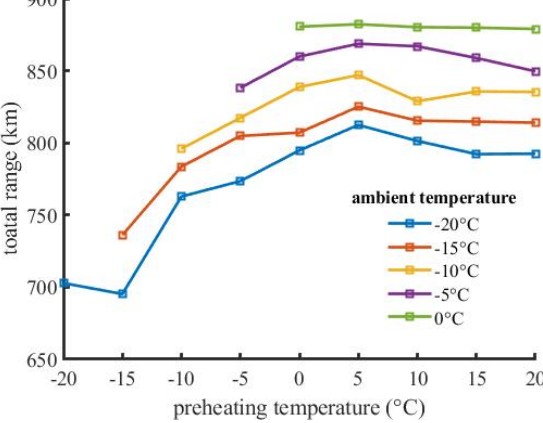

**Figure 10.** Simulation results under the NEDC driving condition.

### 5.3. Heating Strategy under Constant Speed Cruise

To eliminate the different driving characteristics, the next group of simulations was set at a constant speed condition. At different cruising speeds, we compared the driving range with or without preheating. The results are shown in Figure 11, wherein the blue surface represents the results without battery preheating while the red surface represents the results with battery preheating to 5 °C. If the average speed of the next driving process is known (for example, through big data), the preheat/non-preheat driving range under the current state can be compared, thus determining whether preheating is needed.

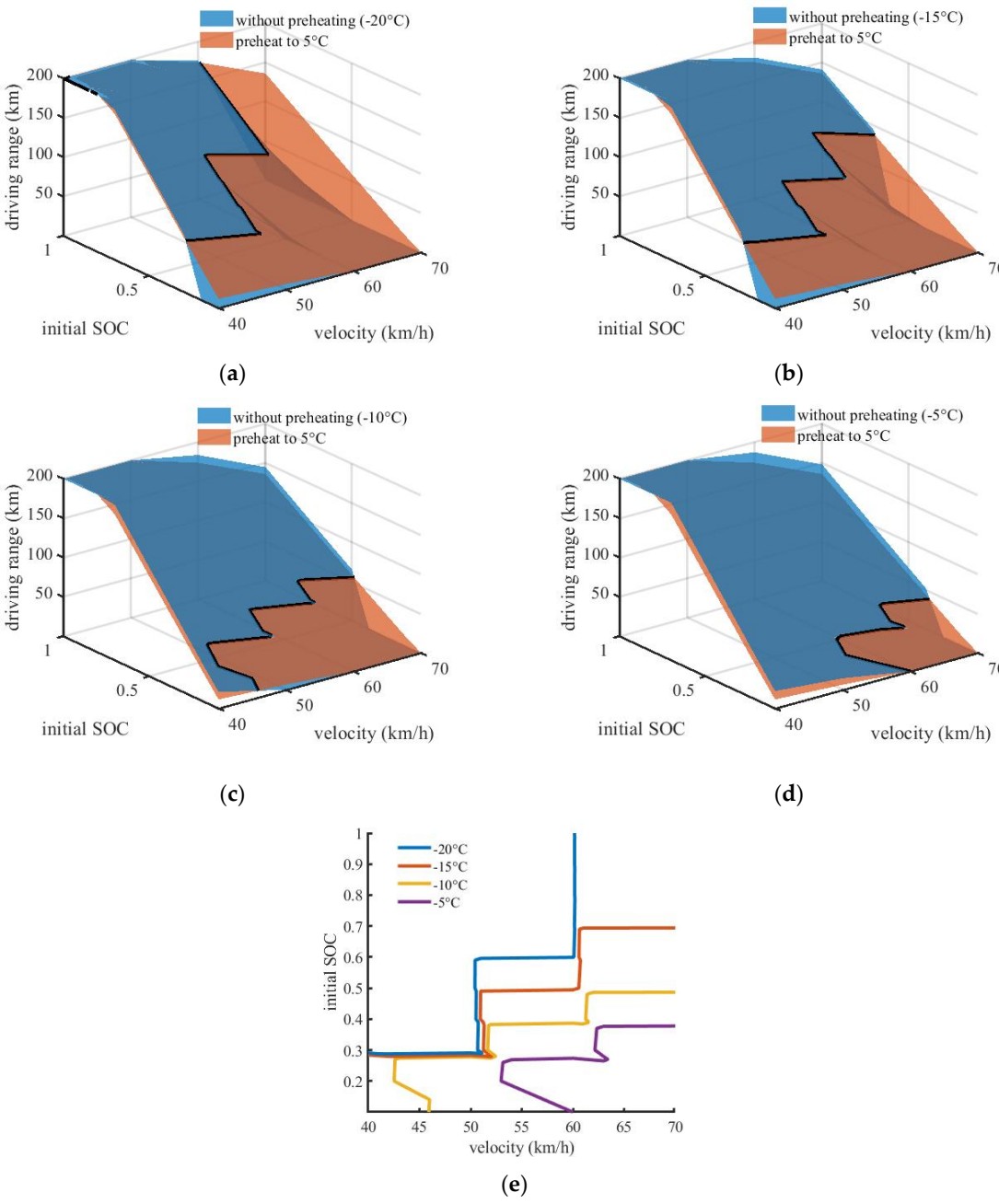

**Figure 11.** Simulation under constant speed driving conditions: (**a**) ambient temperature is −20 °C; (**b**) ambient temperature is −15 °C; (**c**) ambient temperature is −10 °C; (**d**) ambient temperature is −5 °C; and (**e**) heating boundary of each ambient temperature.

Figure 11e is the intersection of the two surfaces in each ambient temperature, representing the boundary of adopting preheating or not. In each ambient temperature, the heating strategy should be chosen if the average speed is in the lower-right corner.

## 6. Conclusions and Suggestions for Improvement

In this paper, an EV model was constructed based on the mass-produced Wuling HongGuang Mini EV. Real vehicle dynamic driving data was used to validate the model. The model was simulated and the reason for attenuation was evaluated quantitatively. Results show that battery energy loss and breaking recovery energy loss contribute nearly half of the range attenuation, which may be alleviated by battery preheating. Future simulations were conducted and the optimal heating method for increasing driving range has been proposed.

Based on the analysis above, some basic suggestions for improving EV performance include:

1.  Improve air conditioning efficiency (e.g., adopting heat pump technology to reduce energy consumption);
2.  Adopt battery preheating methods to raise its temperature during cold weather. It is a clever choice for most conditions, especially at low SOC and low temperatures, which will reduce battery energy loss and breaking recovery energy loss. Furthermore, both the NEDC/WLTC driving range and high cruising speed driving range will be improved by preheating;
3.  Preheat the battery to about 5 °C if possible. The simulation results show that the best preheating temperature is around 5 °C under most conditions to avoid conflicts and simplify the decision making, and preheating to about 5 °C is a simple heating strategy; and
4.  Improve the battery charging ability at low temperatures to recover more energy.

This study focused on the reason for the range attenuation of EVs during winter and on the performance improvement after battery pre-heating. Thus, a lumped-parameter battery pack model was used and the study only considered the heating effect, while the heating time was not considered. For further research, different heating methods can be compared and both the heating effect and heating time must be considered. To propose a complete heating strategy for vehicle application during winter, a more detailed battery model is needed; each battery should be modeled independently; and the model of external heat sources as well as the related physical processes, such as its effects' convection and the heat transfer coefficient, should be considered. These topics are currently being examined by our research team.

**Author Contributions:** Conceptualization, L.L.; methodology, S.M.; software, M.H.; validation, Y.L. and S.M.; formal analysis, M.H. and S.M.; writing—original draft preparation, S.M.; writing—review and editing, M.H.; supervision, L.L., X.H. and M.O.; funding acquisition, Y.L. and J.S. All authors have read and agreed to the published version of the manuscript.

**Funding:** This work was supported by the International Science & Technology Cooperation Program of China under grant number 2019YFE0100200.

**Data Availability Statement:** The datasets used and/or analyzed during the present study are available from the corresponding author upon request.

**Acknowledgments:** The author thanks Ruipu Energy Co., Ltd.; KeyPower Technologies Co., Ltd.; and SAIC GM Wuling Co., Ltd., for providing technical documentation and data.

**Conflicts of Interest:** The authors declare no conflict of interest. Jie Shao is employee of SAIC GM Wuling Automobile Co., Ltd. and Yong Lu is employee of Beijing KeyPower Technologies Co., Ltd. The paper reflects the views of the scientists, and not the company.

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
