# Peer review of "Analysis and Improvement Measures of Driving Range Attenuation of Electric Vehicles in Winter"

_wevj, doi:10.3390/wevj12040239_

Round 1

Reviewer 1 Report

  1. I suggest the reviewing of the theoretical background that supports the simulation procedure including effects like the battery capacity dependence on the discharge rate
  2. I also recommend focusing the study on the WLTP/WLTC protocol, which is more accurate to real driver’s way of driving than NEDC
  3. Concerning thermal analysis I recommend to include external heat sources as well as including in the mathematical development the effects of convection into the heat transfer coefficient
  4. Improve the quality of the paper following suggested modifications (see comments)

Author Response

We thank the reviewer for careful reading of our manuscript.

For the response of the reviewer's comments, please see the attachment.

Reviewer 2 Report

This paper investigates the driving range attenuation of EVs in winter. The topic is interesting. The operation data collected from an EV is used to model the passenger cabin thermal, battery state and vehicle dynamic. The simulation experiments are carried out to verify the proposed method. The main concerns are listed as follows:

  1. In the Abstract, it is suggested to involve some numerical results from the research to verify the effectiveness of the proposed method.
  2. The contributions of this study are not adequately highlighted in Section 1. For example, how to realize the coupling of the battery, vehicle and driving conditions? Which method is used? Moreover, it is suggested to further give a brief description regrading remaining sections in the end of Section 1.
  3. The equations (2) and (3) are not described in Section 3.1. Meanwhile, some of the notations presented in the equations are not explained.
  4. It is not clear that how to obtain the curves presented in Figure 2. Please give an explanation for them in Section 3.2.1.
  5. It is suggested to give a further explanation regarding the heat production model, as mentioned in Section 3.2.3. What is the main difference between reversible and irreversible heat? Which kind of heat production power is calculated using equation (6)?
  6. In Section 3, it is suggested to further add a subsection to describe the relationships and integration of the models presented in this section. The potential coupling of the different models should be highlighted.
  7. The errors of the estimated results seem to be big as shown in Figure 5. Is the model sufficient to be applied in actual situation? How to ensure its effectiveness? Moreover, is it appropriate to validate the model by only using the data from one day?
  8. It is suggested to give a clear introduction for the energy flow diagram analysis method as mentioned in Section 5.1. Moreover, the Figure 7 should be improved.
  9. In Section 6, it is suggested to give the description regarding the deficiencies and related future works, which is generally presented in the conclusion and useful to further clarify the work.

Author Response

We thank the reviewer for careful reading of our manuscript.

We modified the article point by point and the response to the reviewer's comments are listed below:

  1. The most important numerical result is added in the abstract.
  2. The research methods and steps are described in detail in the penultimate paragraph of the first section. The remaining sections are briefly described in the last paragraph of the first section.
  3. Equation (2) and (3) are described in the article. The notation of some formulas are described in detail in section 3.1.
  4. The identification method of the two parameters in Figure 2 is illustrated in section 3.2.1.
  5. The battery heat generation theory is explained in brief before equation (6) in section 3.2.3.
  6. The construction method of total model is shown in section 4.
  7. The validation process and error analysis are added in section 4.
  8. The energy float diagram is introduced in brief in section 5.1. And the initial Figure 7 is drawn with Microsoft Word, maybe there is some format error, sorry for that and it is replaced by a picture format.
  9. The limitations of this research and further study direction is described in the last section of the article in detail.

We thank reviewer again for the important concern!

Round 2

Reviewer 1 Report

Article has been improved following comments to my first revision; however, I suggest to answer those that still have not been fulfilled completely. Please see the attachment.

Author Response

We appreciate the reviewer for careful reading of our manuscript. The answer and explanation for those that still have not been fulfilled completely are replied in the attached file. Please see the attachment.

Reviewer 2 Report

Thanks for the authors' efforts in their response and revising the paper. I suggest that the manuscript can be accepted for considering of publication after making a little more revision.

  1. The manuscript mentions that the resistance-SOC curve and OCV-SOC curve under different temperatures are obtained by HPPC tests. However, the principle and feature regarding the HPPC test are not discussed. Please give a more detailed description for the HPPC test in Section 3.2.1.
  2. The Figure 7 should be further improved, because the words presented in the figures insert the chart.

Author Response

  1. The authors appreciate the reviewer for comment. The HPPC test is explained in line 170-173. It is a battery test cycle widely used in parameter identification, its source is America FreedomCAR Battery Test Manual For Power-Assist Hybrid Electric Vehicles.

  1. Figure 7 is improved, the new version doesn’t have words insert the chart. Thank the reviewer for suggestion.

Round 3

Reviewer 1 Report

Word "control" in line 112 (last version) should be replaced by "monitor"